# Two-years mothering into the pandemic: Impact of the three COVID-19 waves in the Argentinian postpartum women's mental health

**Agustín Ramiro Miranda**[1]*, **Ana Veronica Scotta**[2,3], **Mariela Valentina Cortez**[2,3], **Elio Andrés Soria**[2,3]

1 MoISA, Univ Montpellier, CIHEAM-IAMM, CIRAD, INRAE, Institut Agro, IRD, Montpellier, France, 2 Universidad Nacional de Córdoba, Facultad de Ciencias Médicas, Córdoba, Argentina, 3 Consejo Nacional de Investigaciones Científicas y Técnicas, INICSA, Córdoba, Argentina

* agustin.miranda@ird.fr

## Abstract

The COVID-19 pandemic disproportionately affects certain vulnerable groups, including postpartum women. Thus, this work aimed to analyze the mental health evolution in Argentinian postpartum women during the first three waves of COVID-19 and its determinants. In this repeated cross-sectional study, data were collected during the three waves of COVID-19: May-July/2020 (n=319), April-August/2021 (n=340), and December/2021-March/2022 (n=341). Postpartum depression, insomnia, and perceived stress symptoms were assessed using valid instruments. Statistical analyses included multivariate logistic regression, analysis of variance, and structural equation modeling to test for temporal trends in mental health indicators during the pandemic and to identify their determinants. The prevalence rates of postpartum depression and insomnia rose from 37% to 60% and 46% to 62%, respectively. In contrast, pandemic-related stress decreased. The following negative factors for maternal mental health were identified: unemployment status, lack of medical support, reduced family size, remote working, advanced maternal age, late postpartum, multiparity, and living in the least developed region of Argentina. Structural equation modeling confirmed a process of pandemic-stress adaptation, although there is a persistent increment of postpartum depression and consequent increased insomnia. Postpartum women's mental health worsened during the COVID-19 pandemic. Although women have become more able to cope and perceive less pandemic-related stress, its social and economic impact still persists and puts them at higher psychological risk. Thus, health systems must seek protection of women of reproductive age against negative factors in order to cope with pandemic-related events.

## Introduction

Since its inception, the COVID-19 pandemic has impacted people's daily lives worldwide [1]. Government measures to mitigate viral transmission, the fear of becoming infected, the loss

**Data availability statement:** The data presented in this study are openly available in Open Science Framework (OSF) at https://osf.io/v69tf/

**Funding:** EAS. Secretaría de Ciencia y Tecnología, Universidad Nacional de Córdoba (grant number SECYT-UNC 273/2020). The funders had no role in study design, data collection and analysis, decision to publish, or preparation of the manuscript.

**Competing interests:** The authors have declared that no competing interests exist.

of loved ones, and the economic impact, among others, have been stressors for psychosocial risk [1]. This critical situation has increased mental disorders, such as depression, anxiety, and insomnia, especially in the most vulnerable populations [2]. In this regard, women during peripartum are particularly susceptible to the negative effects of pandemic stressors. As such, the prevalence of postpartum depression has increased during the pandemic, driven by several contributing factors, including overmedicalization of obstetric health care, fear about getting infected, gender disparity in the allocation of caregiving responsibilities, economic adversities, and social isolation [3,4].

Women face several intrapsychic challenges during the transition to motherhood, characterized by a "new psychic organization" where they redefine their roles and self-perceptions, often experiencing physical and psychological symptoms [5]. These symptoms linked to postpartum depression and anxiety indicate difficulties in adapting to the new roles [5]. Maternal distress-inducing factors also include changes in body, responsibilities, and social circumstances [6,7]. Effective adaptation to motherhood involves internal resources (e.g., self-esteem and coping strategies) and external resources (e.g., social support), which mitigate this burden and protect mental health [7]. Notably, maternal mental health issues also affect their children. In this sense, when caregiver responsiveness is impaired, maternal-child interactions and attachment are disrupted [8,9]. Consequently, children may exhibit poorer social engagement, regulatory behaviors, coping, emotionality, sleep quality, eating behavior, and development [10,11]. Furthermore, long-term outcomes include an increased risk of psychosocial issues in adulthood [11]. Therefore, interventions to address maternal psychological disorders are crucial, as they improve both women and child health, mitigate long-term societal costs, and foster healthier emotional and developmental trajectories for children.

Postpartum depression is one of the most prevalent mental disorders in women after childbirth and usually occurs up to the first year after that, affecting the mother-child-family relationship [12]. Women suffering from postpartum depression may exhibit a persistent low mood, a disinterest or inability to derive pleasure from usual activities, disruptions in sleep and eating patterns, a significant reduction in energy, feelings of worthlessness or guilt, difficulty concentrating, increased irritability and anger, heightened anxiety, and, in severe cases, even thoughts of self-harm or suicide [12]. The global prevalence of postpartum depression before the COVID-19 pandemic was estimated at around 17% [13]. Studies conducted during the pandemic estimated an increase, with meta-analyses reporting prevalence rates between 28 and 34% [3,14]. In Argentina, a study conducted during the first wave of COVID-19 in 2020 reported that 37% of postpartum women were depressed [4]. Moreover, it was associated with the duration of social isolation and linked to insomnia and cognitive dysfunction. Studying this condition is of utmost importance, as it has many negative effects on both child and maternal physical and mental health [12].

On the other hand, the pandemic has modified people's sleep. In this sense, the estimated global prevalence of insomnia within the first year of the pandemic was approximately 44% of the general population [15], while it affected between 34% and 49% of peripartum women in the same period [16,17]. In Argentina, Miranda et al. documented a 46% prevalence in postpartum women during the first wave [4].

Sleep problems and postpartum depression have been triggered by pandemic-related stressors. Some meta-analyses using studies conducted during the first year of the pandemic, estimated a global prevalence of psychological stress between 45 and 50% [15,18]. Concerning peripartum women, the meta-analysis published by Demissie and Bitew reported a 56% prevalence of stress [16]. Furthermore, perceived pandemic stress was associated with a history of mood disorders, pregnancy loss, health problems in their children, economic impact, and fear of contracting COVID-19, among others, in postpartum women from Argentina [19].

Consequently, this work aimed to analyze the mental health evolution in Argentinian postpartum women during the first three waves of COVID-19 and its risk and protective factors. Drawing from the reviewed literature, it was hypothesized that the prevalence of postpartum depression, insomnia, and stress escalated across the COVID-19 pandemic in Argentina, with these trends being influenced by various sociodemographic and health-related factors.

## Materials and methods

### Subjects and design

For this repeated cross-sectional study, self-report online questionnaires were administered to 1,000 postpartum women from Argentina, who participated during the first three outbreaks of COVID-19: first wave (n = 319, from May 11th to July 24th, 2020), the second wave (n = 340, from April 20th to August 17th, 2021), and third wave (n = 341, from December 20th, 2021 to March 10th, 2022). Fig 1 shows the social, economic, political and health characteristics during the first three waves of COVID-19 in Argentina. Women were recruited from public hospitals, private clinics, and online community recruitment. Inclusion criteria were: adult (≥18 years old), living in Argentina, and postpartum (first twelve months). A written consent in the first section of the online survey was given to all participants before filling the questionnaire. This research was approved by the corresponding Research Ethics Committee (registration codes REPIS-3177; REPIS-011), following the Declaration of Helsinki and current legislation. Recruitment campaigns were conducted during the periods of higher infections, in which postpartum women were invited to voluntarily participate in an online survey regarding the impact of the pandemic on their health. The campaigns using social media were achieved by the following three methods: (1) joining and sharing in existing community Facebook groups about maternity and pregnancy, (2) through a campaign directed to postpartum women on Instagram, (3) by sending the online link of the survey through WhatsApp. In addition, women attending health centers for newborn screening were invited to participate by offering them the name of the Facebook or Instagram account, or by providing them with the link to the survey via WhatsApp. These methods enabled snowball sampling where users could like, share, and circulate the social media post among others. There was no paid advertising nor compensation for participation in the research. Some measures were conducted to avoid repeated responses. In this sense, the time-limited nature of puerperium and the absence of any compensation for participating discouraged multiple responses by the same participants. Additionally, a unique identifier was requested before accessing the questionnaire, consisting of the combination of the first two initials of the first and last name and birth date. This identifier served to control the response reliability and potential repetitions. In order to protect sensitive personal information that could compromise anonymity given ethical constraints, no further data about participants' identity was collected. The data used in the present study are available online at https://osf.io/v69tf/.

### Instruments

**Postpartum depression.** The Spanish version of the Postpartum Depression Screening Scale-Short Form (PDSS-SF) is a widely used 7-item scale to assess symptoms suggestive of depressive disorders during the last two weeks: sleep/eating disturbances, anxiety/insecurity, emotional lability, mental confusion, loss of self, guilt/shame, and suicidal thoughts [20]. Participants rated each item using a 5-point Likert scale, ranging from 1 ("strongly disagree") to 5 ("strongly agree"). The total PDSS-SF score varies from 7 to 35, with higher scores suggesting a greater level of postpartum depression. A cut-off score of 17 or greater suggests

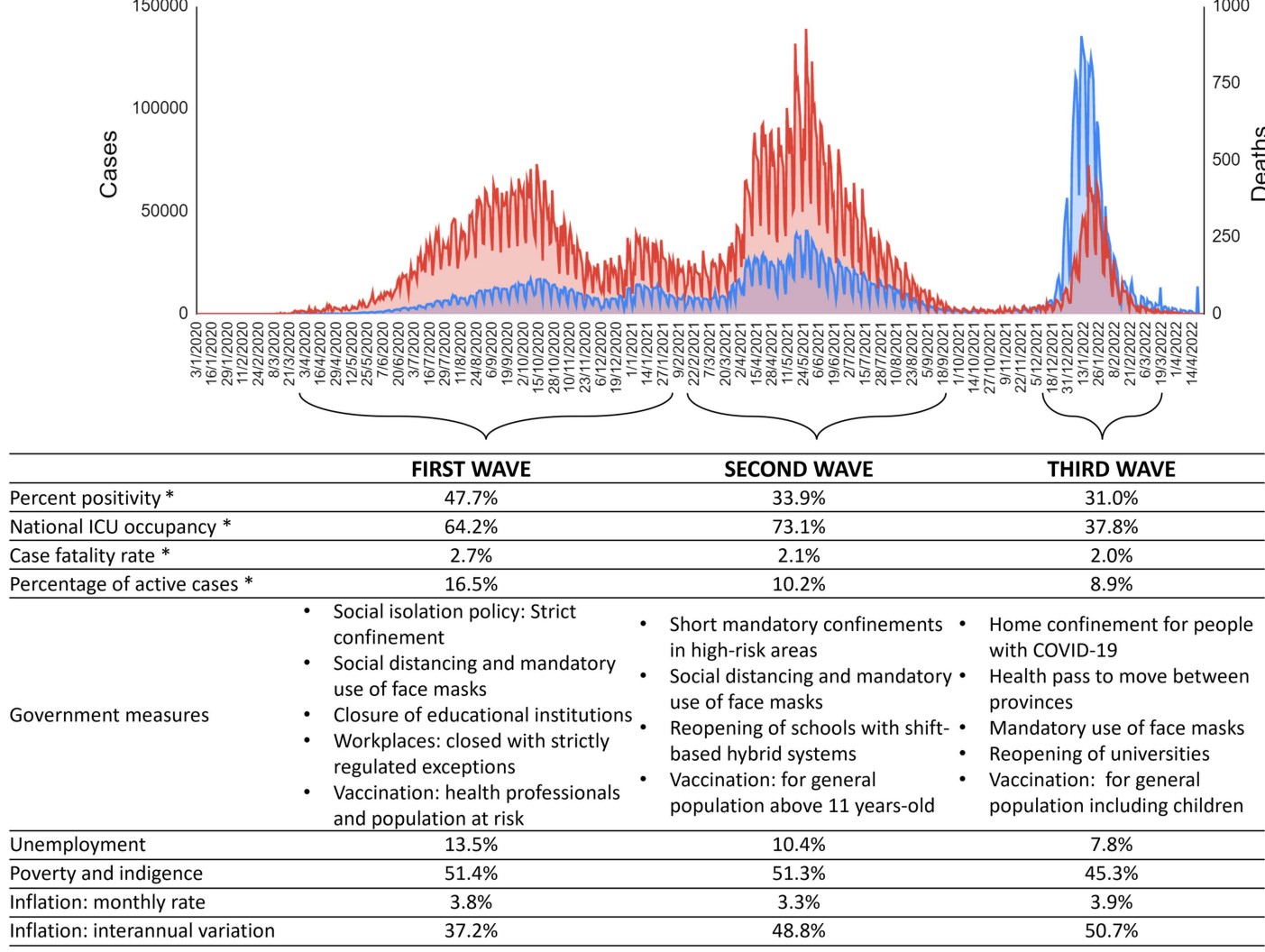

| | FIRST WAVE | SECOND WAVE | THIRD WAVE |
|---|---|---|---|
| Percent positivity * | 47.7% | 33.9% | 31.0% |
| National ICU occupancy * | 64.2% | 73.1% | 37.8% |
| Case fatality rate * | 2.7% | 2.1% | 2.0% |
| Percentage of active cases * | 16.5% | 10.2% | 8.9% |
| Government measures | • Social isolation policy: Strict confinement<br>• Social distancing and mandatory use of face masks<br>• Closure of educational institutions<br>• Workplaces: closed with strictly regulated exceptions<br>• Vaccination: health professionals and population at risk | • Short mandatory confinements in high-risk areas<br>• Social distancing and mandatory use of face masks<br>• Reopening of schools with shift-based hybrid systems<br>• Vaccination: for general population above 11 years-old | • Home confinement for people with COVID-19<br>• Health pass to move between provinces<br>• Mandatory use of face masks<br>• Reopening of universities<br>• Vaccination: for general population including children |
| Unemployment | 13.5% | 10.4% | 7.8% |
| Poverty and indigence | 51.4% | 51.3% | 45.3% |
| Inflation: monthly rate | 3.8% | 3.3% | 3.9% |
| Inflation: interannual variation | 37.2% | 48.8% | 50.7% |

**Fig 1. Social, economic, political and health characteristics during the first three waves of COVID-19 in Argentina.** * Estimates of the epidemiological week with the most cases during the wave are presented, according to the report of the Ministry of Health of Argentina. Socio-economic statistics were taken from the national Institute of Statistics and Census of Argentina.

a clinically significant level of depression (89% sensitivity and 77% specificity) [4]. PDSS-SF showed an acceptable reliability (alpha = 0.841).

**Insomnia.** The Spanish version of the Insomnia Severity Index (ISI) is a questionnaire comprising seven items designed to evaluate the nature, severity, and impact of insomnia during the previous month. This traditional tool encompasses the subsequent aspects: severity of sleep onset, sleep maintenance, early morning awakening problems, sleep dissatisfaction, interference of sleep difficulties with daytime functioning, noticeability of sleep problems by others, and distress caused by sleep difficulties [21]. Each item is rated on a 5-point Likert scale, ranging from 0 to 4, where a score of 0 indicates the absence of any sleep-related concern ("no problem") while a score of 4 corresponds to a very severe problem. The items are summed to result in total

scores ranging from 0 to 28, with higher scores indicating more severe insomnia and a higher impact on life quality. A cut-off of ≥ 10 was used to identify clinically significant symptomatology, with a sensitivity of 86% and a specificity of 88% [4,21]. ISI exhibited adequate reliability (alpha = 0.822).

**Perceived stress.** The Spanish version of the COVID-19 Pandemic-related Perceived Stress Scale (PSS-C) is a 10-item scale assessing the extent to which pandemic-related situations are appraised as stressful during the past month [22]. The participants rated each item on a 5-point Likert scale, with 0 meaning they never experienced the described condition and 4 meaning they experienced it very often. The total score, ranging from 0 to 40, reflected the level of perceived stress, with higher scores indicating higher stress levels. PSS-C has adequate psychometric properties and a bidimensional structure. The Stress factor focuses on the perceived lack of control over the situation (items 1, 2, 3, 6, 9, and 10), while the Coping factor is centered on the cognitive and behavioral strategies required to deal with stress-inducing circumstances (items 4, 5, 7, and 8). As this second factor involved inverse items, higher scores reflect low perceived ability to cope with pandemic stressors. A cut-off of ≥ 20 was used to identify stressed women (sensitivity of 83% and specificity of 61%) [23]. Reliability in the current sample for the entire PSS-C and its subscales were α = 0.823, 0.821, and 0.664, respectively.

**Sociodemographic and health characteristics.** Maternal age was dichotomised as < 35 years old and ≥ 35 years-old, as it is associated with obstetric complications that can lead to mental health disorders [4]. Participants gave information about the partnership status (single or in a couple) and the number of cohabitants (number of people with whom she is isolated without counting herself). Regarding educational level, it was classified into two groups: < 12 years and ≥ 12 years of formal education, which is the compulsory minimum according to Argentine law (i.e., completed secondary education). They also reported whether they took online classes (yes or no). Concerning work, women were classified as employed or unemployed, and whether they performed remote work (yes or no). The number of days since the beginning of COVID-19 in Argentina to the time of answering the questionnaire was calculated for each participant.

Furthermore, it was registered if they had health insurance (yes or no), the mode of delivery (vaginal or Cesarean), parity (primiparous or multiparous), postpartum period (<6 months or ≥ 6 months), gestational age at delivery (term or preterm), and history of pregnancy loss (yes or no). Regarding breastfeeding, women's adherence to World Health Organization's recommendations was considered (yes or no) if they comply with exclusive breastfeeding from 0 to 6 months or mixed lactation from 6 to 12 months. To categorize this variable, women were asked about the type of feeding at the time of answering the question-naire (exclusive breastfeeding, mixed, or artificial -formula-). Also, women reported if they received medical support on breastfeeding (yes or no).

Regarding geographical location, women were grouped in three regions of Argentina [24]. The asymmetry in the level of development among Argentina's provinces makes it possible to distinguish three main regions: Core including Ciudad Autónoma de Buenos, Aires, Buenos Aires, Cordoba, and Santa Fe (they contain almost two-thirds of the national population and three-quarters of the gross domestic product), North including Jujuy, Salta, Formosa, Chaco, Tucumán, Santiago del Estero, Catamarca, La Rioja, San Juan, San Luis, Mendoza, Misiones, Corrientes, and Entre Ríos (they have the lowest per capita incomes with the highest national rates of social deterioration), and South including La Pampa, Neuquén, Río Negro, Chubut, Santa Cruz, and Tierra del Fuego Antártida e Islas del Atlántico Sur (they exhibit low population density but a higher socioecomic support per capita with a better living standard) [25].

## Statistical analysis

Statistical analyses were conducted using Stata software (version 18, StataCorp, College Station, TX, USA), and p values below 0.05 were considered significant. Descriptive statistics (mean and standard deviation) were calculated for all numerical variables; the percentage of categorical variables was also described. The reliability was measured by the Cronbach's alpha coefficient for internal consistency.

Multivariate logistic regression was performed to compare the prevalence of mental health conditions among COVID-19 waves and related factors. For this purpose, questionnaire scores were categorized according to cut-off points differentiating between women with and without symptoms of depression, insomnia and postpartum stress. Results were expressed as odds ratios (OR), confidence interval (IC95%), and p values. Also, multivariate regressions were conducted to identify determinant factors of symptoms of postpartum depression and insomnia, reporting adjusted β coefficients and p values. In this regard, the scalar scores of the items of the postpartum depression and insomnia questionnaires were used, since each item represents a specific symptom. Similarly, the relationship between the duration of pandemic and mental health status was assessed using multivariate regressions to establish its evolution over time. In addition, analysis of variance was used to compare means of each instrument among waves, using the Bonferroni *post-hoc test*. Levene's test was used to analyze homogeneity of variance, whilst skewness (S) and kurtosis (K) were used to test variable distributions. Multicollinearity between the independent variables included in the models was checked by calculating the variance inflation factor (VIF) and the tetrachoric correlation coefficients. Problems of multicollinearity were considered if two variables had a correlation $\geq 0.80$, or the mean VIF was $\geq 6$, or the highest individual VIF was $\geq 10$. Tetrachoric correlations and VIF of all variables suggested no significant multicollinearity (S1 and S2 Figs).

Finally, structural equation modeling (SEM) was used to examine the effects of COVID-19 pandemic on postpartum mental health. The maximum likelihood estimation was adopted to estimate the model fit, with the calculation of goodness-of-fit indices: Chi-square to degree of freedom ratio ($\chi 2/df$), root mean square error of approximation (RMSEA), comparative fit index (CFI), Tucker Lewis index (TLI), and standardized root mean square residual (SRMR). The minimum sample size to detect effects in the final SEM model with 2 latent variables and 17 observed variables was 411, assuming an anticipated effect size of 0.15, desired statistical power level of 0.80, and probability level of 0.05. Thus, the current sample is deemed suitable for analyzing the proposed model.

## Results

Most women were < 35 years old (76%) and had completed $\geq 12$ years of formal education (95%). Regarding location, 67% of them lived in the Core region of Argentina, 17% in the North, and 16% in the South. Regarding their working status, 75% were employed, and 24% were remote workers. Additionally, 34% of participants took online courses. Respecting their health, 87% had health insurance, 38% were multiparous, 83% of births were in term, and 63% were delivered by Cesarean section. Also, 46% were $\geq 6$ months postpartum, and 25% reported a history of pregnancy loss. Regarding lactation, 69% adhered adequately to WHO's breastfeeding recommendations, and 62% received medical support. Finally, 95% of the sample were in a couple, 51% resided at least with 2 people, 40% with 2 to 5, and 9% with more than five. Table 1 displays the evolution of the prevalence of symptoms of mental disorders during the COVID-19 pandemic.

The women's symptoms of postpartum depression and its conditioning factors are displayed in Table 2. All symptoms increased significantly across the three waves of COVID-19.

**Table 1. Prevalence of symptoms of mental disorders during the COVID-19 pandemic in postpartum women in Argentina.**

| | Prevalence of mental symptoms % (n) | | | | Mean (SD) and testing of assumptions of scale scores | | | | |
|---|---|---|---|---|---|---|---|---|---|
| | Total | First wave | Second wave | Third wave | Scale score | Levene test | P | S | K |
| PDSS-SF | 47.30 (473) | 36.68 (117) | 44.41 (151) | 60.12 (205) | 16.84 (6.94) | 1.5337 | 0.2160 | 0.56 | -0.52 |
| ISI | 55.80 (558) | 46.39 (148) | 58.53 (199) | 61.88 (211) | 10.51 (5.68) | 0.1803 | 0.8351 | 0.22 | -0.41 |
| PSS-C | 29.81 (298) | – | 37.65 (128) | 21.99 (75) | 15.85 (6.48) | 0.9756 | 0.3236 | 0.09 | -0.25 |

PDSS-SF = Postpartum Depression Screening Scale-Short Form (≥ 17 = symptoms suggestive of postpartum depression); ISI = Insomnia Severity Index (≥10 = symptoms suggestive of insomnia); PSS-C = COVID-19 Pandemic-related Perceived Stress Scale (≥20 = symptoms suggestive of psychological stress); SD = standard deviation, S = skewness; K = kurtosis. Levene's test to test homogeneity of variance across waves.

Employment was the main factor associated inversely with postpartum depression. On the other hand, remote working was associated with an increase in suicidal thoughts ($\beta = 0.07$, $p = 0.032$). Sleep disturbances were positively associated with multiparity ($\beta = 0.09$, $p = 0.012$), ≥6 months of postpartum ($\beta = 0.08$, $p = 0.008$), and history of pregnancy loss ($\beta = 0.07$, $p = 0.022$), which was also associated with mental confusion ($\beta = 0.09$, $p = 0.007$). Having 3-4 cohabitants was inversely related to sleep disturbances ($\beta = -0.08$, $p = 0.023$), anxiety/insecurity ($\beta = -0.09$, $p = 0.009$), and mental confusion ($\beta = -0.07$, $p = 0.035$). In addition, having a couple decreased anxiety/insecurity ($\beta = -0.08$, $p = 0.015$) and suicidal thoughts ($\beta = -0.13$, $p < 0.001$). Conversely, women who lived in the North region showed significantly more suicidal thoughts than those in the Core and South regions. Although adhering to the WHO's recommendations for breastfeeding was not associated with any depression symptoms, receiving medical support significantly decreased anxiety/insecurity, emotional lability, mental confusion, and feeling guilt/shame.

On the other hand, Table 3 presents the symptoms of postpartum insomnia and its conditioning factors. All concerns increased significantly across the COVID-19 waves, except for noticeability. Studying for ≥12 years, being employed, being in a couple, living with 3-4 cohabitants, and having medical support were significant protective factors against insomnia. On the contrary, symptoms were worsened by ≥6 postpartum months, being ≥35 years, multiparity, Cesarean delivery, working remotely, having a history of pregnancy loss, and living in the North region of Argentina.

Prevalences of postpartum depression, insomnia, and stress in the total sample were 47.30%, 55.80%, and 29.81%, respectively. When each COVID-19 wave was analyzed, the prevalence of postpartum depression was 36.68% in the first wave, 44.41% in the second wave, and 60.12% in the third wave. Fig 2 shows the results of multivariate logistic regression. The risk of having symptoms of postpartum depression was consecutively increased across waves. Those women who were employed had a 38% lower risk of postpartum depression (OR = 0.62, SE = 0.11, $p = 0.006$), although working remotely from home tended to increase the risk (OR = 1.37, SE = 0.22, $p = 0.054$). In addition, women who lived with 3 or 4 people had a lower risk than those who lived with fewer people (OR = 0.72, SE = 0.11, $p = 0.027$). Medical support during lactation reduced the risk of postpartum depression (OR = 0.65, SE = 0.09, $p = 0.002$).

Concerning insomnia, the prevalence was 46.39% in the first wave, 58.53% in the second wave, and 61.88% in the third wave. As shown in Fig 2, women during the second and third waves were at greater risk of insomnia than women during the first one (OR = 1.50, SE = 0.26, $p = 0.019$; and OR = 1.78, SE = 0.30, $p = 0.001$, respectively). Women older than 35 years (OR = 1.45, SE = 0.24, $p = 0.024$), after 6 months of postpartum (OR = 1.39, SE = 0.19, $p = 0.015$),

**Table 2. Multivariate regression models for symptoms of postpartum depression across COVID-19 waves according to Argentinian women characteristics (n = 1,000).**

| | Sleep disturbances | Anxiety/ Insecurity | Emotional lability | Mental confusion | Loss of self | Guilt/ Shame | Suicidal thoughts |
|---|---|---|---|---|---|---|---|
| Wave | | | | | | | |
| Second vs. first | −0.01 (p = 0.859) | 0.12 (p = 0.003) | 0.04 (p = 0.334) | 0.13 (p < 0.001) | 0.07 (p = 0.091) | 0.07 (p = 0.090) | 0.04 (p = 0.255) |
| Third vs. first | 0.14 (p < 0.001) | 0.18 (p < 0.001) | 0.15 (p < 0.001) | 0.22 (p < 0.001) | 0.22 (p < 0.001) | 0.14 (p < 0.001) | 0.16 (p < 0.001) |
| Third vs. second | 0.15 (p < 0.001) | 0.07 (p = 0.076) | 0.11 (p = 0.006) | 0.09 (p = 0.017) | 0.16 (p < 0.001) | 0.08 (p = 0.057) | 0.12 (p = 0.003) |
| ≥ 35 years-old | 0.01 (p = 0.732) | 0.01 (p = 0.997) | 0.00 (p = 0.929) | 0.06 (p = 0.073) | 0.00 (p = 0.986) | −0.01 (p = 0.865) | −0.02 (p = 0.626) |
| ≥ 6 months of postpartum | 0.08 (p = 0.008) | 0.05 (p = 0.092) | −0.04 (p = 0.249) | 0.02 (p = 0.443) | 0.04 (p = 0.220) | 0.02 (p = 0.597) | 0.03 (p = 0.368) |
| Multiparity | 0.09 (p = 0.012) | −0.02 (p = 0.512) | 0.00 (p = 0.908) | 0.05 (p = 0.161) | 0.01 (p = 0.894) | 0.05 (p = 0.149) | 0.06 (p = 0.087) |
| Adherence to WHO's breastfeeding recommendation | 0.02 (p = 0.574) | 0.01 (p = 0.685) | −0.05 (p = 0.136) | −0.02 (p = 0.528) | −0.01 (p = 0.748) | −0.05 (p = 0.158) | −0.01 (p = 0.666) |
| Cesarean delivery | 0.02 (p = 0.569) | 0.03 (p = 0.391) | 0.03 (p = 0.332) | −0.02 (p = 0.602) | −0.01 (p = 0.844) | 0.00 (p = 0.920) | 0.05 (p = 0.118) |
| ≥ 12 years of formal education | 0.02 (p = 0.499) | −0.02 (p = 0.499) | −0.05 (p = 0.109) | 0.05 (p = 0.109) | 0.02 (p = 0.549) | −0.01 (p = 0.718) | −0.02 (p = 0.470) |
| Online learning | 0.02 (p = 0.474) | 0.04 (0.180) | 0.01 (p = 0.728) | 0.05 (p = 0.084) | −0.03 (p = 0.342) | −0.01 (p = 0.865) | −0.03 (p = 0.304) |
| Employment | −0.07 (p = 0.044) | −0.12 (p < 0.001) | −0.11 (p = 0.002) | −0.13 (p < 0.001) | −0.11 (p = 0.001) | −0.07 (p = 0.050) | −0.12 (p = 0.001) |
| Remote working | 0.06 (p = 0.064) | −0.13 (p < 0.001) | 0.02 (p = 0.538) | 0.03 (p = 0.402) | 0.05 (p = 0.154) | 0.00 (p = 0.933) | 0.07 (0.032) |
| In couple | 0.03 (p = 0.293) | −0.08 (p = 0.015) | −0.06 (p = 0.077) | 0.01 (p = 0.664) | 0.03 (p = 0.370) | −0.02 (p = 0.631) | −0.13 (p < 0.001) |
| Medical insurance | −0.04 (p = 0.283) | −0.01 (p = 0.744) | −0.02 (p = 0.498) | 0.02 (p = 0.525) | −0.03 (p = 0.402) | −0.08 (p = 0.021) | −0.03 (p = 0.342) |
| History of pregnancy loss | 0.07 (p = 0.022) | 0.04 (p = 0.207) | 0.03 (p = 0.380) | 0.09 (p = 0.007) | 0.03 (p = 0.418) | 0.02 (p = 0.583) | 0.02 (p = 0.575) |
| Full-term pregnancy | −0.01 (p = 0.805) | 0.02 (p = 0.451) | −0.01 (p = 0.782) | −0.01 (p = 0.867) | 0.01 (p = 0.816) | −0.01 (p = 0.848) | −0.04 (p = 0.165) |
| 3–4 cohabitants | −0.08 (p = 0.023) | −0.09 (p = 0.009) | −0.03 (p = 0.457) | −0.07 (p = 0.035) | −0.05 (p = 0.150) | −0.01 (p = 0.877) | 0.00 (p = 0.911) |
| 5 or more cohabitants | −0.01 (p = 0.803) | −0.07 (p = 0.033) | −0.02 (p = 0.504) | −0.01 (p = 0.805) | −0.02 (p = 0.582) | 0.01 (p = 0.765) | −0.01 (p = 0.680) |
| Region | | | | | | | |
| North vs. Core | 0.05 (p = 0.146) | 0.03 (p = 0.334) | 0.06 (p = 0.100) | 0.02 (p = 0.565) | 0.01 (p = 0.814) | 0.04 (p = 0.281) | 0.11 (p = 0.002) |
| South vs. Core | 0.01 (p = 0.765) | −0.01 (p = 0.758) | 0.01 (p = 0.832) | −0.02 (p = 0.593) | −0.01 (p = 0.651) | 0.01 (p = 0.675) | −0.02 (p = 0.598) |
| North vs. South | 0.04 (p = 0.360) | 0.04 (p = 0.318) | 0.05 (p = 0.257) | 0.04 (p = 0.387) | 0.02 (0.594) | 0.02 (p = 0.597) | 0.12 (p = 0.004) |
| Medical support | −0.06 (p = 0.077) | −0.13 (p < 0.001) | −0.07 (p = 0.031) | −0.09 (p = 0.003) | −0.04 (p = 0.235) | −0.09 (p = 0.007) | −0.05 (p = 0.152) |

Data presented as beta coefficients and p-values.

**Table 3. Symptoms of postpartum insomnia across COVID-19 waves according to Argentinian women characteristics (n = 1,000).**

| | Onset | Maintenance | Awakening | Dissatisfaction | Interference | Noticeability | Distress |
|---|---|---|---|---|---|---|---|
| Wave | | | | | | | |
| Second vs. first | 0.06 (p = 0.141) | 0.06 (p = 0.115) | 0.09 (p = 0.031) | 0.01 (p = 0.798) | 0.06 (p = 0.159) | 0.04 (p = 0.277) | 0.05 (p = 0.183) |
| Third vs. first | 0.13 (p = 0.001) | 0.11 (p = 0.005) | 0.10 (p = 0.012) | 0.11 (p = 0.004) | 0.14 (p < 0.001) | 0.02 (p = 0.588) | 0.14 (p < 0.001) |
| Third vs. second | 0.07 (p = 0.058) | 0.05 (p = 0.242) | 0.01 (p = 0.772) | 0.10 (p = 0.011) | 0.08 (p = 0.039) | −0.02 (p = 0.576) | 0.09 (p = 0.025) |
| ≥ 35 years-old | −0.01 (p = 0.858) | 0.04 (p = 0.222) | 0.02 (p = 0.480) | 0.07 (p = 0.036) | 0.02 (p = 0.633) | 0.01 (p = 0.848) | 0.06 (p = 0.097) |
| ≥ 6 months of postpartum | 0.14 (p < 0.001) | 0.11 (p = 0.001) | 0.05 (p = 0.162) | 0.12 (p < 0.001) | 0.06 (p = 0.051) | −0.01 (p = 0.834) | 0.09 (p = 0.004) |
| Multiparity | 0.04 (p = 0.254) | 0.05 (p = 0.182) | 0.06 (p = 0.092) | 0.11 (p = 0.002) | 0.14 (p < 0.001) | 0.06 (p = 0.108) | 0.06 (p = 0.065) |
| Adherence to WHO's breastfeeding recommendation | 0.01 (p = 0.747) | 0.02 (p = 0.579) | 0.00 (p = 0.974) | 0.02 (p = 0.570) | −0.03 (p = 0.326) | 0.01 (p = 0.814) | 0.01 (p = 0.820) |
| Cesarean delivery | 0.07 (p = 0.021) | 0.04 (p = 0.202) | 0.03 (p = 0.381) | 0.02 (p = 0.454) | 0.05 (p = 0.151) | 0.03 (p = 0.415) | 0.02 (p = 0.550) |
| ≥ 12 years of formal education | −0.09 (p = 0.006) | −0.06 (p = 0.072) | −0.07 (p = 0.031) | 0.02 (p = 0.538) | −0.01 (p = 0.843) | 0.00 (p = 0.984) | −0.03 (p = 0.457) |
| Online learning | 0.04 (p = 0.262) | 0.04 (p = 0.187) | 0.06 (p = 0.071) | −0.02 (p = 0.500) | 0.00 (p = 0.988) | −0.03 (p = 0.290) | 0.02 (p = 0.552) |
| Employment | −0.11 (p = 0.002) | −0.04 (p = 0.217) | −0.04 (p = 0.243) | −0.01 (p = 0.849) | −0.02 (p = 0.612) | 0.01 (p = 0.826) | −0.04 (p = 0.268) |
| Remote working | 0.06 (p = 0.070) | 0.03 (p = 0.429) | 0.00 (p = 0.995) | 0.02 (p = 0.557) | 0.07 (p = 0.027) | −0.01 (p = 0.757) | 0.08 (p = 0.018) |
| In couple | −0.10 (p = 0.001) | −0.01 (p = 0.697) | 0.01 (p = 0.677) | 0.04 (p = 0.203) | 0.04 (p = 0.200) | −0.06 (p = 0.090) | 0.01 (p = 0.805) |
| Medical insurance | 0.02 (p = 0.492) | 0.05 (p = 0.163) | −0.01 (p = 0.829) | 0.03 (p = 0.377) | 0.02 (p = 0.585) | 0.07 (p = 0.051) | 0.06 (p = 0.104) |
| History of pregnancy loss | 0.06 (p = 0.077) | 0.07 (p = 0.043) | 0.03 (p = 0.365) | 0.08 (p = 0.008) | 0.02 (p = 0.489) | −0.07 (p = 0.043) | 0.03 (p = 0.368) |
| Full-term pregnancy | −0.06 (p = 0.054) | −0.03 (p = 0.335) | −0.06 (p = 0.052) | −0.03 (p = 0.296) | 0.01 (p = 0.838) | −0.03 (p = 0.289) | −0.02 (p = 0.516) |
| 3-4 cohabitants | −0.05 (p = 0.141) | −0.08 (p = 0.026) | −0.01 (p = 0.757) | −0.11 (p = 0.002) | −0.09 (p = 0.013) | 0.01 (p = 0.841) | −0.07 (p = 0.043) |
| 5 or more cohabitants | −0.01 (p = 0.845) | −0.04 (p = 0.279) | 0.02 (p = 0.582) | −0.02 (p = 0.527) | 0.05 (p = 0.156) | −0.01 (p = 0.863) | 0.00 (p = 0.951) |
| Region | | | | | | | |
| North vs. Core | 0.01 (p = 0.701) | 0.02 (p = 0.611) | 0.02 (p = 0.601) | 0.08 (p = 0.013) | 0.06 (p = 0.061) | 0.04 (p = 0.235) | 0.05 (p = 0.184) |
| South vs. Core | 0.00 (p = 0.899) | 0.03 (p = 0.319) | −0.02 (p = 0.534) | −0.01 (p = 0.729) | 0.02 (p = 0.610) | −0.06 (p = 0.062) | −0.04 (p = 0.239) |
| North vs. South | 0.01 (p = 0.837) | −0.02 (p = 0.715) | 0.04 (p = 0.373) | 0.10 (p = 0.026) | 0.05 (p = 0.279) | 0.10 (p = 0.018) | 0.08 (p = 0.051) |
| Medical support | −0.03 (p = 0.362) | −0.03 (p = 0.323) | −0.02 (p = 0.598) | −0.09 (p = 0.004) | −0.04 (p = 0.226) | 0.05 (p = 0.148) | −0.02 (p = 0.651) |

Data presented as beta coefficients and p-values.

and multiparous (OR = 1.60, SE = 0.24, p = 0.002) were more likely to have postpartum insomnia. Also, women who lived with 3 or 4 people had lower risks than those who lived with fewer people (OR = 0.73, SE = 0.11, p = 0.037). Additionally, women living in the North region of Argentina had a higher risk of insomnia with respect to those living in the South.

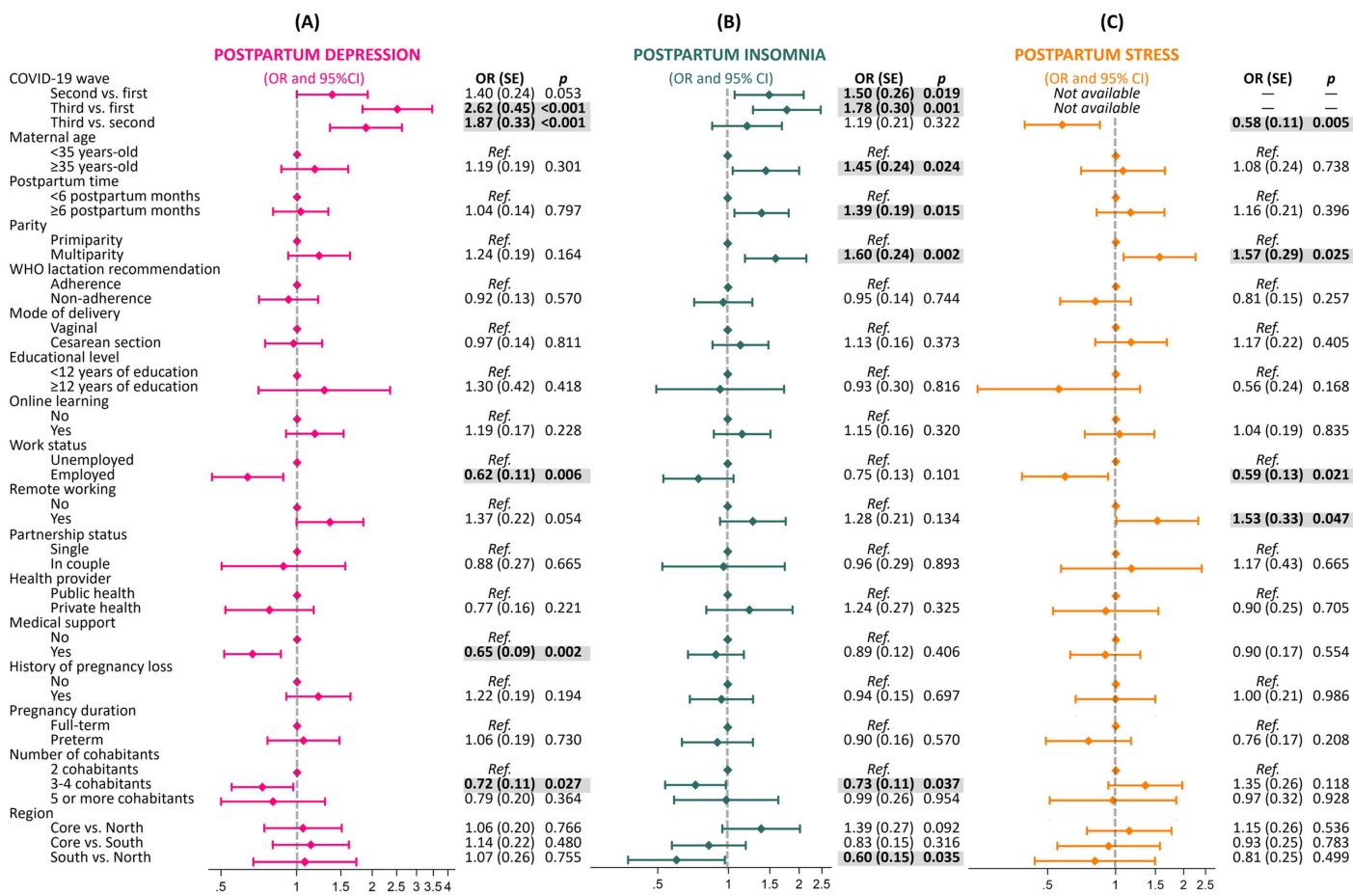

**Fig 2. Logistic regression plot of adjusted odds ratios (OR) and 95% confidence intervals (CI) for postpartum depression, insomnia and stress across the three first waves of COVID-19 pandemic in Argentina (n = 1,000).**

Pandemic stress decreased from 37.65% in the second wave to 21.99% in the third wave. Furthermore, logistic regression revealed that women in the third wave were at lower risk than those in the second one (OR = 0.58, SE = 0.11, p = 0.005) (Fig 2). Multiparity was associated with an increased risk of stress (OR = 1.57, SE = 0.29, p = 0.025). Similarly, women who were working from home were at higher stress risk OR = 1.53, SE = 0.33, p = 0.047). Employment was a protective factor against pandemic stress (OR = 0.59, SE = 0.13, p = 0.021).

Fig 3A shows added-variable plots of the multivariate regressions when pandemic duration is included as a continuous variable. It was found that the total scores of PDSS and ISI increased significantly as the pandemic progressed (β = 0.21, p < 0.001; and β = 0.13, p < 0.001, respectively). On the contrary, the days of pandemic were inversely associated with PSS total score (β = −0.19, p < 0.001), as well as its subscales: Coping (β = −0.11, p = 0.013) and Stress (β = −0.18, p < 0.001).

Fig 3B shows the ANOVA comparisons of total scores in the questionnaires according to the COVID-19 waves. First, Levene's test confirmed the homogeneity of variance among groups (Table 1). With respect to PDSS, the mean scores were 15.04 (SD = 6.42, S = 0.81, K = −0.22) in the first wave, 16.71 (SD = 6.91, S = 0.56, K = −0.64) in the second wave, and 18.66 (SD = 7.01, S = 0.38, K = −0.50) in the third one, with significant differences among them. Regarding the ISI scores, the respective means were: 9.42 (SD = 5.49, S = 0.37,

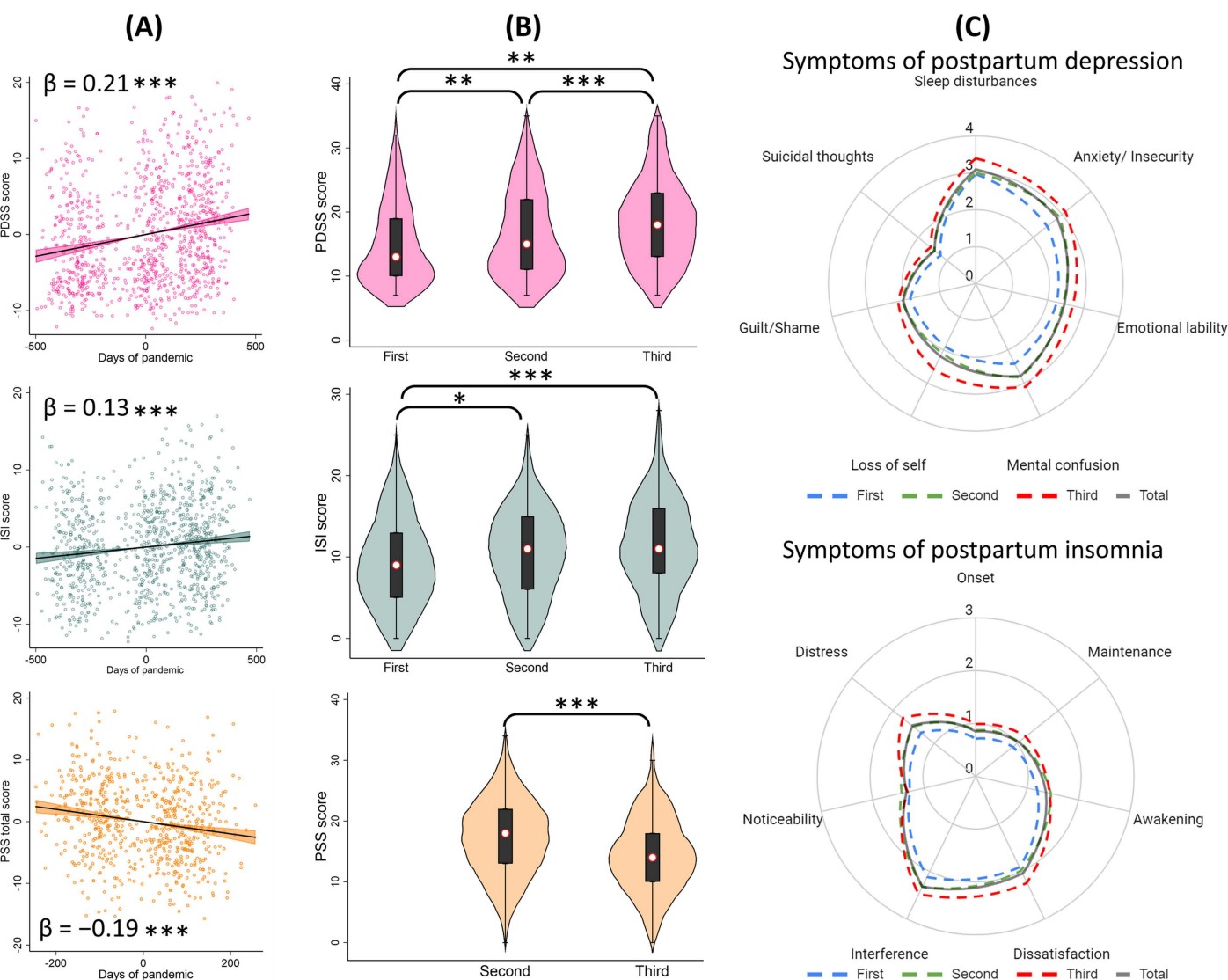

**Fig 3. Association between postpartum mental health and the duration of the COVID-19 pandemic in Argentina and comparisons between waves (n = 1,000).** Note: results derived from multivariate regressions (a) and analyses of variance (b). Radar charts (c) displaying mean scores of symptoms included in the instruments. * p < 0.05; ** p < 0.01; *** p < 0.001.

K = −0.44), 10.61 (SD = 5.63, S = 0.10, K = −0.46), and 11.44 (SD = 5.73, S = 0.18, K = −0.30). ANOVA showed statistical differences when comparing the first wave versus the second and third ones. The mean PSS total score was 17.19 (SD = 6.48, S = −0.05, K = −0.20) in the second wave and 14.47 (SD = 6.21, S = 0.19, K = −0.14) in the third one, which were significantly different. Increased radar chart surfaces were obtained according to COVID-19 waves, indicating higher mean scores for all postpartum depression and insomnia symptoms over time (Fig 3C).

Finally, Fig 4 presents the SEM results for changes in maternal mental health status during the COVID-19 pandemic. Model I revealed that the days of pandemics had positive direct effects on postpartum depression (βs = 0.23, p < 0.001) but not on postpartum insomnia, which in turn was directly affected by postpartum depression (βs = 0.57, p < 0.001). When including pandemic stress in the model (Model II), it showed that days of the pandemic had

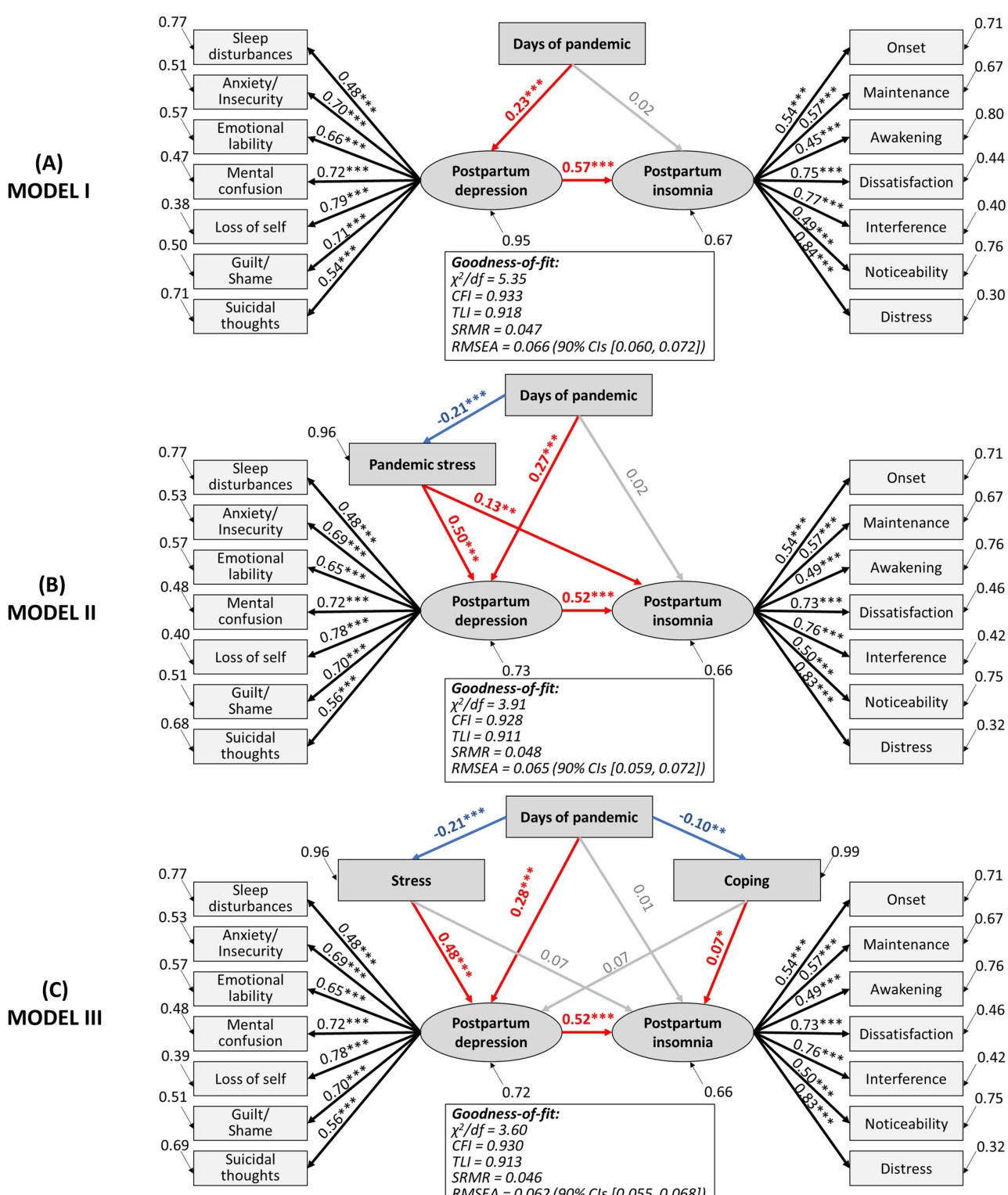

**Fig 4. Structural equation model of potential predictors of maternal mental health of postpartum women during the COVID-19 pandemic in Argentina.** Standardized paths coefficients are displayed. Ellipses are used to denote latent constructs, and rectangles are used to denote the observed variables. Red arrows indicate positive direct effects, blue arrows indicate negative direct effects, and gray arrows indicate non-significant relationships. $\chi^2$/df = chi-square value relative to the degrees of freedom; CFI=comparative fit index; TLI=Tucker Lewis index; SRMR=standardized root mean square residual; RMSEA=root mean square error of approximation; CIs = confidence interval.

a negative effect on stress (βs = −0.21, p < 0.001). Nonetheless, pandemic stress had direct effects on postpartum depression (βs = 0.50, p < 0.001) and, to a lesser extent, on postpartum insomnia (βs = 0.13, p < 0.01). On the other hand, Model III divides pandemic stress into the coping and stress subscales. This final model confirms that the duration of the pandemic had positive direct effects on postpartum depression (βs = 0.28, p < 0.001) and negative on distress (βs = −0.21, p < 0.001) and coping (βs = −0.10, p < 0.01), whilst distress had positive effects on depression (βs = 0.48, p < 0.001) and coping on insomnia (βs = 0.07, p < 0.05). Also, postpartum depression had positive effects on insomnia (βs = 0.52, p < 0.001). Results indicated that the model-data fit was excellent for the three models.

## Discussion

This study aimed to analyze the evolution of the prevalence of postpartum depression, insomnia, and perceived pandemic stress in Argentinian women during the first three waves of COVID-19, as well as to identify risk and protective factors of puerperal mental health. Our results showed that postpartum depression and insomnia increased over time. Employment status, medical support for breastfeeding, and living with family members were factors that reduced the risk of postpartum depression, whereas remote work increased it. The most prevalent symptoms of postpartum depression were sleep disorders, anxiety/insecurity, mental confusion, and emotional lability, which also increased as the pandemic progressed.

Regarding insomnia, the risk factors were maternal age, late postpartum, multiparity, and living in the North region of Argentina, whereas family size was a protective factor. The main insomnia problem was sleep maintenance, and the most prevalent complaints were daytime interference, dissatisfaction, and insomnia-related distress. On the other hand, pandemic stress decreased over time, with multiparous, remote workers and unemployed women having a higher risk of stress. Comprehensively, SEM models demonstrated that postpartum depression increased during the first two years of the COVID-19 pandemic, with a consequent rise in insomnia. Moreover, women adapted to pandemic stressors over time, which impaired puerperal mental health.

Postpartum depression is one of the most prevalent mental disorders after childbirth. Studies before the COVID-19 pandemic report prevalence rates of around 10% in high-income countries, 20% in middle-income countries, and 25% in low-income countries. In this study, we found that postpartum depression increased significantly across the three waves of COVID-19 in Argentina, which rose from 37% in the first wave to 60% in the third wave. These results are consistent with previous research demonstrating higher prevalences of postpartum depression during pandemic phases with higher numbers of infections. For example, Citu et al. [26] found that those women who gave birth during the fourth pandemic wave had a higher rate of depression than those from the first wave, being characterized by a difference of more than 17%. Similarly, Babicki et al. [27] studied postpartum mental health during the first, second, and third waves, and found that anxiety and depressive symptoms increased as the pandemic progressed. An increase in maternal depression, anxiety, and stress in pregnant women was reported during the second wave compared to the first wave in Canada, although a decrease in all scores was observed during the third wave [28]. A Swedish study reported an impairment of depressive symptoms and discomfort during the first wave, followed by a normalization during the following months and a further increase when cases began to rise again (second wave) [29]. Moreover, Bajaj et al. [30] documented a significant rise in postpartum depressive symptoms within the first year following the onset of the COVID-19 pandemic in the United States, suggesting a potential link between the pandemic and increased postpartum depression.

Over time, the pandemic has increasingly negatively affected the mental health of the general population. This impact worsened during periods of higher COVID-19 cases and stricter public health measures, indicating that prolonged and severe pandemic conditions place greater strain on people's mental well-being. In this sense, Fernández et al. [31] and del-Valle et al. [32] found that psychological distress, including depression, increased significantly over time, with the highest levels in 2020 (first wave) and 2021 (second wave) in Argentina. Fernández et al. [31] reported that the temporal variations in mental status were heterogeneous: symptoms of pandemic stress decreased in phases with higher caseloads and more restrictive measures, while there was an increase in mood symptoms, negative affect, difficulties in social relationships, and impulse regulation. In addition, del-Valle et al. [32] showed a significant time-related effect. The sustained increase in depression during the pandemic in Argentina is consistent with findings from other countries. The prevalence of depressive symptoms ranged from 28% in 2020 to 33% in 2021 in the United States [33], from 20% in the first lockdown in 2020 to 25% in the second lockdown in 2021 in Austria [34], and from 16% to 18% in Belgium [35].

Over the past two years, it has become clear that the pandemic has exacerbated many pre-existing gender inequalities at work and at home, which increase the risk for female psychological disorders [36]. In this regard, women are more vulnerable to job losses, job casualization, and increased demands for care at home [36]. In addition, during the reopening of economies, Latin American women lag behind men in recovering their jobs [37]. Consistently with the literature, the current study showed that employed women had a lower risk of postpartum depression, and employment was negatively associated with all symptoms of depression. Alfayumi-Zeadna et al. [38] found that during the pandemic in Israel, the likelihood of perinatal depression symptoms experienced by unemployed women was 2.7 times higher than that for employed women. Another study conducted in China found that being unemployed increased the odds of having depression in pregnant and new mothers by 34% [39]. Moreover, in the present work, employed women were 41% less likely to be stressed by the COVID-19 pandemic. Similarly, Mollard et al. [40] and Thapa et al. [41] reported that PSS-10 scores were lower in postpartum women with employment. Nonetheless, women who were working from home were 1.53 times more likely to be stressed compared to those working on-site. This finding is consistent with previous research reporting that women in remote work are more vulnerable to stress than men during the COVID-19 pandemic [42]. In addition, remote working may be more challenging for mothers due to overlapping work and childcare responsibilities [43]. Further, our results showed that multiparous women were 57% more likely to have pandemic stress. Therefore, work-from-home policies must address gender inequalities to ensure the well-being of working mothers.

The increase in suicidal thoughts found in this study is a relevant aspect to highlight. This finding is consistent with prior research showing an increased occurrence of suicidal ideation among postpartum women during the COVID-19 pandemic [44]. We found that this severe symptom of postpartum depression was related to several sociodemographic factors: unemployment, single-partnership status, residence in the North region of Argentina, and remote work. Suicidal ideation during peripartum has been associated with the economic impact of the COVID-19 pandemic [45], unemployment [46], singleness [47], and living in a low-income region [48]. Furthermore, Lin et al. [49] studied adults from the United States during the first pandemic phase to find a positive correlation between pandemic fears and suicidal ideation, which fully accounted for insomnia severity.

Analyses showed that medical support in breastfeeding practice was a protective factor against depression. Those women who received support from medical personnel were 35% less likely to have postpartum depression. This is consistent with the study by Ostacoli et

al. [50], who found that perceived support from the healthcare staff was a protective factor against postpartum depression and posttraumatic stress symptoms. Breastfeeding support is essential for maternal mental health, as women who experience negative support have an increased risk for postpartum depression [51]. During the pandemic, breastfeeding support has declined. According to Brown and Shenker, 27% of women had difficulty obtaining adequate breastfeeding support during COVID-19 [52]. Similarly, more than 35% of women did not receive breastfeeding support during the first wave in Argentina [4]. Health systems are encouraged to strengthen breastfeeding support and resources during social and health crises to improve the mental well-being of postpartum women. Numerous innovative solutions have emerged, such as telehealth lifestyle interventions, telelactation, and technology-based prenatal breastfeeding education [53].

Social support is a multidimensional construct that consists of the perceived availability of assistance or comfort from others, being a resource that helps to cope with biological, psychological, and social stressors [54]. Social support may arise from any interpersonal relationship in a social network (e.g., family, friends, neighbors, and caregivers) and is a key factor in mental well-being during the postpartum period [55]. According to Fiorillo et al. [56], protective factors against the development of psychiatric symptoms during the pandemic include higher levels of satisfaction with life and cohabiting people, and living with a greater number of family members. In the current study, we found that women who cohabited with 3 or 4 people had a lower risk of postpartum depression and insomnia than those who cohabited with up to two people. A previous study demonstrated that the postpartum women's family social network (cohabitants) during social isolation in Argentina was a protective factor that reduced the risk of postpartum depression [4]. Similar findings were reported by Gustafsson et al. [57], who showed that women with severe and persistent postpartum depression reported weaker social support, including the number of family members. Perceived social support, including support from family, has been linked to a lower likelihood of experiencing moderate to high levels of anxiety, depression, and stress among pregnant women during the COVID-19 pandemic [58]. Thus, this support might function as a protective factor for the mental well-being of peripartum women during this challenging period. In another study conducted by Tsuno et al. [59], it was observed that postpartum depression in Japanese women was linked to two primary factors: the social restrictions imposed during the COVID-19 pandemic and a reduction in social support from healthcare providers, family, and friends. However, household chaos during the pandemic, which was greater in those families with a single caregiver or large family size, promoted parental depression and anxiety [60]. It is well established that household composition (i.e., family size, living with dependent children, having outdoor space) has played an important role in mental health in people during the lockdowns, affecting mood, sleep quality, and stress levels [61]. According to our results and the literature, multiparous women had a higher risk of insomnia and stress when compared with primiparous ones [62,63]. Likewise, we found that in addition to multiparity, advanced maternal age, months of puerperium, and living in the North region of Argentina were insomnia promoters, coinciding with the postpartum sleep determinants documented by Khadka et al [64].

On the other hand, the prevalence of insomnia showed an upward trend similar to that of depression: 46.39%, 58.53%, and 61.88%. Other studies have shown an increase in insomnia and depression throughout the pandemic in different population groups, being more severe when lockdowns were implemented [65,66]. Stay-at-home measures were associated with altered sleep schedules, quality, patterns, and duration due to multifactorial causes, such as loss of social synchronizers, reduced physical activity, increased stress and anxiety, and intrafamily conflicts [67]. Moreover, women are more prone to have pandemic-related insomnia than men, which may be explained by gender gaps [68]. Concerning this, women spend more

time on family duties, such as homeschooling, unpaid housework, nurturing, and caregiving [68]. Thus, peripartum women are even more vulnerable to sleep disturbances during this pandemic, with prevalences of insomnia ranging from 34% to 75% [4,16,69].

In contrast to depression and insomnia, our findings indicate that pandemic-related stress decreased from 38% to 22%. These results are consistent with other studies confirming an adaptation to pandemic stressors over time [70]. In addition, more flexible governmental restrictions, better knowledge about the infection, and availability of vaccines, among others, favor coping with pandemic stressors. In this regard, Mullins et al. found that levels of concern about COVID-19 decline over time, which would be related to the notion of "*back to normal*" and vaccination campaigns [71]. Similar results are reported by Rogowska et al., who found a significant reduction in perceived stress [72]. Regarding peripartum women, Demissie and Bitew [16] reported a pooled prevalence of 56%, but higher rates have been documented during the earliest pandemic phases [40].

Structural equation modeling showed that the pandemic duration had a direct effect on postpartum depression but not insomnia. Rather, insomnia was directly influenced by postpartum depression, which agrees with the role of depression as a predictor of insomnia in the general population and peripartum women during COVID-19 [4,73]. Also, Casagrande et al. [74] reported a model demonstrating that social distancing and stay-at-home measures lead to psychological disorders that worsen sleep quality in the population. Therefore, the treatment of insomnia during this pandemic would benefit from incorporating treatments for depression. Likewise, there is also a need to prioritize insomnia treatment and sleep hygiene when treating depression, as they have a bidirectional relationship.

Moreover, when pandemic stress was included in our model, it showed a declining trend, but promoted both postpartum depression and insomnia. These results indicate that although postpartum women perceive lower levels of pandemic stress as the pandemic progresses, which is consistent with previous studies [70,71] stressed women are more susceptible to depression and insomnia. Pandemic-related stress has been strongly associated with peripartum depressive symptoms during the pandemic [38,75]. Moreover, Anderson et al. [76] analyzed the experiences of pregnant and postpartum women with a history of depressive symptoms, finding that pandemic-related worries and fears, as well as drastic changes in their work and household responsibilities and governmental restrictions, placed them at risk for worsening their mental well-being. Our results showed that pandemic-related distress is reduced and coping skills against pandemic stressors are improved over time. In addition, stress was found to have a direct effect on postpartum depression, while lack of coping skills promoted insomnia. It has been evidenced that phenotypes of pregnant and postpartum women with high levels of active coping strategies (e.g., social support and self-care) are associated with higher resilience, whereas passive coping (e.g., increased screen time and social media use) was associated with elevated symptoms of depression, anxiety, global psychological distress, and sleep problems during the COVID-19 pandemic [77]. Coinciding with Raman et al. [78], the relationship between postpartum depression and insomnia persisted when COVID-19 related-stress was introduced into the model. These authors suggest that the observed reciprocal temporal relationships are not trivial, and that depression and insomnia are mutually dependent constructs that share a dependent relationship with the anxiety specifically related to concerns about the pandemic (COVID-19 anxiety).

The results of the present study show that postpartum women have increased their ability to cope with pandemic stressors during the course of the pandemic, leading to a lower level of perceived stress. However, this adaptive process to pandemic stressors has not necessarily involved a reduction in postpartum depression or insomnia. Our data suggest that the COVID-19-related socioeconomic impact is a key factor in mental well-being, and it is necessary to implement health strategies and policies that protect peripartum women as a socially

vulnerable group. The pandemic has shown a dynamic behavior and its evolution must be analyzed according to the particularities of each country. In the case of Argentina (see Fig 1), when analyzing the official indicators reported by the Ministry of Health [79], the second wave was the one that presented the peak with the highest number of deaths (case fatality rate of 2.1%), while the third wave was the peak of the highest positive cases (percentage of positivity 31%). However, although some indicators such as the percentage of ICU occupancy improved from the second to the third wave, most did not show such marked changes. This includes socio-economic indicators published by the government, where a negative impact on the economy can be observed due to a sustained increase in inflation [80]. However, during the last wave there was an improvement in the unemployment rate and in the number of people living in poverty and indigence, although they continue to represent a significant social burden. Therefore, each model in the present study was adjusted by the wave and socioeconomic factors in order to have accurate and reliable results, considering that it is not a static process, hence the need to make these multiple adjustments. Consequently, there was a temporal adaptation of the constraints used in the corresponding models.

These findings should be interpreted within the limitations of our data. First, the repeated cross-sectional study design, rather than longitudinal, did not allow us to assess individual changes over time. However, this subtype of cross-sectional study, also known as serial or pseudo-longitudinal, is a useful method for analyzing population changes over time (also known as aggregate change over time) [81]. Despite that, interpretations derived from the aggregation of data from different waves may be affected by the social context and specific characteristics of each of the three subsamples. These factors included events such as the changes in health policies related to the pandemic over the study duration [82]. Nonetheless, the multivariate statistical modeling performed herein considers the pandemic waves as potential confounders to obtain reliable and precise estimates. Also, it is necessary to emphasize that, although repeated cross-sectional studies are inappropriate to observe individual-level changes, they are adequate to assess population-level changes [83]. Second, as part of our exploration of the evolution of pandemic-related stress, the absence of PSS-C data during the first wave of COVID-19 represents a potential limitation. This gap arose because the Spanish PSS-C adaptation for the pandemic context was published after data collection for our initial sample had been completed [84]. Furthermore, since the lack of data affected all participants in this wave, rather than a specific subset, we opted not to impute the missing values in order to prevent potential bias. Considering the exceptional circumstances of the first wave, which was marked by unprecedented levels of uncertainty and stress for all, and recognizing that this phase tended to be the most stressful [85], the absence of data during this period is not expected to substantially affect the overall conclusions of the study. Third, we lack evidence of prior psychiatric diagnoses, domestic violence, and consumption of psychotropic drugs, that have been proposed as determinants of postpartum mental health. Fourth, mental health has been assessed by self-report instruments in this study, thus future work needs to include objective measures of depression (e.g., clinical diagnosis), insomnia (e.g., actigraphy), and stress (e.g., hormone dosage). Despite these limitations, our findings broaden knowledge about the impact of the current COVID-19 pandemic on the mental health of postpartum women, and are useful for designing strategies and making decisions in future pandemics or crisis scenarios.

## Conclusion

In summary, our results demonstrate that the mental health of Argentinian postpartum women deteriorated during the COVID-19 pandemic. Our research findings suggest a notable increase in the prevalence of postpartum depression and insomnia during the outbreaks.

Moreover, the social and economic impact involved a high psychological risk, although women have become able to cope with pandemic-related stress. Thus, these findings emphasize the critical importance of routine mental health assessments during perinatal visits, especially in future similar epidemiologic scenarios, to address the risk of postpartum depression, stress, and insomnia for vulnerable mothers.

## Supporting information

**S1 Fig. Tetrachoric correlation matrix between variables.**
(TIF)

**S2 Fig. Variance Inflation Factor (VIF). Dash line denotes the mean VIF.**
(TIF)

## Author contributions

**Conceptualization:** Agustin Ramiro Miranda, Ana Veronica Scotta, Mariela Valentina Cortez, Elio Andrés Soria.

**Data curation:** Agustin Ramiro Miranda, Ana Veronica Scotta.

**Formal analysis:** Agustin Ramiro Miranda.

**Funding acquisition:** Elio Andrés Soria.

**Investigation:** Agustin Ramiro Miranda, Ana Veronica Scotta.

**Methodology:** Agustin Ramiro Miranda, Ana Veronica Scotta, Mariela Valentina Cortez.

**Project administration:** Agustin Ramiro Miranda, Mariela Valentina Cortez.

**Resources:** Agustin Ramiro Miranda, Elio Andrés Soria.

**Software:** Agustin Ramiro Miranda.

**Supervision:** Agustin Ramiro Miranda.

**Validation:** Agustin Ramiro Miranda.

**Visualization:** Agustin Ramiro Miranda.

**Writing – original draft:** Agustin Ramiro Miranda, Ana Veronica Scotta, Mariela Valentina Cortez, Elio Andrés Soria.

**Writing – review & editing:** Agustin Ramiro Miranda, Ana Veronica Scotta, Mariela Valentina Cortez, Elio Andrés Soria.

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
