## [Decision Letter · Decision Letter 0]

11 Jun 2024

PONE-D-23-34221Two-years mothering into the pandemic: Impact of the three COVID-19 waves in the Argentinian postpartum women’s mental healthPLOS ONE

Dear Dr. Miranda,

Thank you for submitting your manuscript to PLOS ONE. After careful consideration, we feel that it has merit but does not fully meet PLOS ONE’s publication criteria as it currently stands. Therefore, we invite you to submit a revised version of the manuscript that addresses the points raised during the review process.

We look forward to receiving your revised manuscript.

Kind regards,

Daniel Ahorsu, PhD

Academic Editor

PLOS ONE

Journal Requirements:

2. You indicated that ethical approval was not necessary for your study. We understand that the framework for ethical oversight requirements for studies of this type may differ depending on the setting and we would appreciate some further clarification regarding your research. Could you please provide further details on why your study is exempt from the need for approval and confirmation from your institutional review board or research ethics committee (e.g., in the form of a letter or email correspondence) that ethics review was not necessary for this study? Please include a copy of the correspondence as an ""Other"" file.

4. Thank you for stating the following financial disclosure: 'EAS. Secretaría de Ciencia y Tecnología, Universidad Nacional de Córdoba (grant number SECYT-UNC 273/2020)."  

5. In the online submission form, you indicated that data contain private health information and cannot be shared publicly. The minimal dataset will be made available upon request for researchers who meet the criteria for access to confidential data. Interested researchers should contact the corresponding author on agustin.miranda@ird.fr

6. We note that Figure 2 in your submission contain [map/satellite] images which may be copyrighted. All PLOS content is published under the Creative Commons Attribution License (CC BY 4.0), which means that the manuscript, images, and Supporting Information files will be freely available online, and any third party is permitted to access, download, copy, distribute, and use these materials in any way, even commercially, with proper attribution. For these reasons, we cannot publish previously copyrighted maps or satellite images created using proprietary data, such as Google software (Google Maps, Street View, and Earth). For more information, see our copyright guidelines: http://journals.plos.org/plosone/s/licenses-and-copyright.

Additional Editor Comments:

Dear Authors,

It's been great going through your manuscript. The amount of time and effort put into collecting data in this difficult period is appreciable. However, I have two minor comments: 1) How do you take care of participants who responds to the questionnaire twice or more taking into consideration that it took about two years to gather the data (3 waves); and 2) what are the limitations to using data of three waves spanning over two years especially as all the data were combined too (n=1000).

Reviewers' comments:

Reviewer's Responses to Questions

**Comments to the Author**

1. Is the manuscript technically sound, and do the data support the conclusions?

Reviewer #1: Yes

Reviewer #2: Yes

2. Has the statistical analysis been performed appropriately and rigorously? 

Reviewer #1: Yes

Reviewer #2: Yes

3. Have the authors made all data underlying the findings in their manuscript fully available?

Reviewer #1: No

Reviewer #2: Yes

4. Is the manuscript presented in an intelligible fashion and written in standard English?

Reviewer #1: Yes

Reviewer #2: Yes

5. Review Comments to the Author

Reviewer #1: Thank you very much for the possibility to review the study titled "Two-years mothering into the pandemic: Impact of the three COVID-19 waves in the Argentinian postpartum women’s mental health".

The paper is very interesting and well written. There are only small points to review that could improve the work. In particular, although the introduction and discussions address the issue of the pandemic and the impact it can have very well, it would be important to be able to better discuss these aspects with the difficulties that women experience on an intrapsychic level in the transition to motherhood. In particular, it is suggested to see some studies in this regard:

- Tambelli, R., Ballarotto, G., Trumello, C., & Babore, A. (2022). Transition to Motherhood: A Study on the Association between Somatic Symptoms during Pregnancy and Post-Partum Anxiety and Depression Symptoms. International Journal of Environmental Research and Public Health, 19(19), 12861.

- Ben-Ari, O. T., Shlomo, S. B., Sivan, E., & Dolizki, M. (2009). The transition to motherhood—A time for growth. Journal of Social and Clinical Psychology, 28(8), 943-970.

- Rallis, S., Skouteris, H., McCabe, M., & Milgrom, J. (2014). The transition to motherhood: towards a broader understanding of perinatal distress. Women and Birth, 27(1), 68-71.

Furthermore, the paper partly takes for granted the importance of assessment and prevention in this specific age group. Although there is a very large literature that has highlighted the outcomes in children with mothers suffering from post-partum depression, I think it is important to highlight it, both in the introduction and more importantly in the conclusions and clinical implications. For example, several studies have highlighted the effects both in early childhood and later in children, highlighting the importance of interventions. For example:

- Ballarotto, G., Murray, L., Bozicevic, L., Marzilli, E., Cerniglia, L., Cimino, S., & Tambelli, R. (2023). Parental sensitivity to toddler’s need for autonomy: An empirical study on mother-toddler and father-toddler interactions during feeding and play. Infant Behavior and Development, 73, 101892.

- Liu, X., Wang, S., & Wang, G. (2022). Prevalence and risk factors of postpartum depression in women: a systematic review and meta‐analysis. Journal of Clinical Nursing, 31(19-20), 2665-2677.

- Hutchens, B. F., & Kearney, J. (2020). Risk factors for postpartum depression: an umbrella review. Journal of midwifery & women's health, 65(1), 96-108.

- Cimino, S., Cerniglia, L., Tambelli, R., Ballarotto, G., Erriu, M., Paciello, M., ... & Koren-Karie, N. (2020). Dialogues about emotional events between mothers with anxiety, depression, anorexia nervosa, and no diagnosis and their children. Parenting, 20(1), 69-82.

Abstract.

Acronyms for tools are used in the abstract. Authors are invited to eliminate acronyms. Specifically: "Postpartum depression (PDSS‐SF), insomnia (ISI), and perceived stress symptoms27 (PSS‐C) were used"

Finally, the authors are invited to better clarify the hypotheses of the study, justifying them with the literature

Reviewer #2: The authors have done a good job with the organization of their article in general. To begin with, the research has been dealt with the technical aspect very well, and the conclusion has a very strong link that makes the article a very good one. In addition, the statistical aspect of the article is presented in a professional manner, although I have not gotten access to the original data to examine it. Furthermore, the authors have indicated that the data are available without restriction. Finally, the article presents the write-up in professional and appropriate English grammar throughout the article. I do recommend this article without any competing interest for publication.

6. PLOS authors have the option to publish the peer review history of their article (what does this mean? ). If published, this will include your full peer review and any attached files.

**Do you want your identity to be public for this peer review?** For information about this choice, including consent withdrawal, please see our Privacy Policy .

Reviewer #1: No

Reviewer #2: **Yes: ** Baba Yahaya Mohammed

---

## [Author Response · Author response to Decision Letter 1]

19 Jun 2024

Dear Editor,

We appreciate the valuable suggestions provided by the reviewers and yourself, which have greatly enhanced our work. Consequently, each change made is indicated in the following list:

Editor’s comments:

Response: The manuscript was formatted following the indications in the template provided by the Editor.

2. You indicated that ethical approval was not necessary for your study. We understand that the framework for ethical oversight requirements for studies of this type may differ depending on the setting and we would appreciate some further clarification regarding your research. Could you please provide further details on why your study is exempt from the need for approval and confirmation from your institutional review board or research ethics committee (e.g., in the form of a letter or email correspondence) that ethics review was not necessary for this study? Please include a copy of the correspondence as an ""Other"" file.

Response: As described in the Materials and Methods section, this study had the corresponding ethical approval. “This research was approved by the corresponding Research Ethics Committee (registration codes REPIS-3177; REPIS-011), following the Declaration of Helsinki and current legislation”.

Response: The data presented in this study are openly available in Open Science Framework (OSF) at https://osf.io/v69tf/

4. Thank you for stating the following financial disclosure: 'EAS. Secretaría de Ciencia y Tecnología, Universidad Nacional de Córdoba (grant number SECYT-UNC 273/2020)."

Response: The following statement was added "The funders had no role in study design, data collection and analysis, decision to publish, or preparation of the manuscript."

5. In the online submission form, you indicated that data contain private health information and cannot be shared publicly. The minimal dataset will be made available upon request for researchers who meet the criteria for access to confidential data. Interested researchers should contact the corresponding author on agustin.miranda@ird.fr

Response: The data presented in this study are openly available in Open Science Framework (OSF) at https://osf.io/v69tf/

6. We note that Figure 2 in your submission contain [map/satellite] images which may be copyrighted. All PLOS content is published under the Creative Commons Attribution License (CC BY 4.0), which means that the manuscript, images, and Supporting Information files will be freely available online, and any third party is permitted to access, download, copy, distribute, and use these materials in any way, even commercially, with proper attribution. For these reasons, we cannot publish previously copyrighted maps or satellite images created using proprietary data, such as Google software (Google Maps, Street View, and Earth). For more information, see our copyright guidelines: http://journals.plos.org/plosone/s/licenses-and-copyright.

We require you to either (1) present written permission from the copyright holder to publish these figures specifically under the CC BY 4.0 license, or (2) remove the figures from your submission.

Response: The map was enhanced and made in Stata using open license data. We include the following statement: “This map was created with the spmap package in Stata 18 by the authors, using shapefiles from the open license resource geoBoundaries Global Administrative Database (www.geoboundaries.org) [25].”

Response: Done.

Response: We confirmed that there were no retractions among the references used in the current study. Only one reference (Rogowska et al., 2021) had a corrigendum issued after publication, which corrected an error in the figure captions. This correction did not affect the study's conclusions, and the main document has been updated accordingly.

Additional Editor Comments:

1. Comment: How do you take care of participants who respond to the questionnaire twice or more taking into consideration that it took about two years to gather the data (3 waves). Response: We added the following rationale in Materials and methods:

“Some measures were conducted to avoid repeated responses. In this sense, the time-limited nature of puerperium and the absence of any compensation for participating discouraged multiple responses by the same participants. Additionally, a unique identifier was requested before accessing the questionnaire, consisting of the combination of the first two initials of the first and last name and birth date. This identifier served to control the response reliability and potential repetitions. In order to protect sensitive personal information that could compromise anonymity given ethical constraints, no further data about participants’ identity was collected.”

2. Comment: What are the limitations to using data of three waves spanning over two years especially as all the data were combined too (n=1000):

The following rationale was included in Discussion:

“Despite that, interpretations derived from the aggregation of data from different waves may be affected by the social context and specific characteristics of each of the three subsamples. These factors included events such as the changes in health policies related to the pandemic over the study duration [82]. Nonetheless, the multivariate statistical modeling performed herein considers the pandemic waves as potential confounders to obtain reliable and precise estimates. Also, it is necessary to emphasize that, although repeated cross-sectional studies are inappropriate to observe individual-level changes, they are adequate to assess population-level changes [83].”

Reviewer #1: Thank you very much for the possibility to review the study titled "Two-years mothering into the pandemic: Impact of the three COVID-19 waves in the Argentinian postpartum women’s mental health".

The paper is very interesting and well written. There are only small points to review that could improve the work. In particular, although the introduction and discussions address the issue of the pandemic and the impact it can have very well,

1. Comment: It would be important to be able to better discuss these aspects with the difficulties that women experience on an intrapsychic level in the transition to motherhood. In particular, it is suggested to see some studies in this regard:

- Tambelli, R., Ballarotto, G., Trumello, C., & Babore, A. (2022). Transition to Motherhood: A Study on the Association between Somatic Symptoms during Pregnancy and Post-Partum Anxiety and Depression Symptoms. International Journal of Environmental Research and Public Health, 19(19), 12861.

- Ben-Ari, O. T., Shlomo, S. B., Sivan, E., & Dolizki, M. (2009). The transition to motherhood—A time for growth. Journal of Social and Clinical Psychology, 28(8), 943-970.

- Rallis, S., Skouteris, H., McCabe, M., & Milgrom, J. (2014). The transition to motherhood: towards a broader understanding of perinatal distress. Women and Birth, 27(1), 68-71.

Furthermore, the paper partly takes for granted the importance of assessment and prevention in this specific age group. Although there is a very large literature that has highlighted the outcomes in children with mothers suffering from post-partum depression, I think it is important to highlight it, both in the introduction and more importantly in the conclusions and clinical implications. For example, several studies have highlighted the effects both in early childhood and later in children, highlighting the importance of interventions. For example:

- Ballarotto, G., Murray, L., Bozicevic, L., Marzilli, E., Cerniglia, L., Cimino, S., & Tambelli, R. (2023). Parental sensitivity to toddler’s need for autonomy: An empirical study on mother-toddler and father-toddler interactions during feeding and play. Infant Behavior and Development, 73, 101892.

- Liu, X., Wang, S., & Wang, G. (2022). Prevalence and risk factors of postpartum depression in women: a systematic review and meta‐analysis. Journal of Clinical Nursing, 31(19-20), 2665-2677.

- Hutchens, B. F., & Kearney, J. (2020). Risk factors for postpartum depression: an umbrella review. Journal of midwifery & women's health, 65(1), 96-108.

- Cimino, S., Cerniglia, L., Tambelli, R., Ballarotto, G., Erriu, M., Paciello, M., ... & Koren-Karie, N. (2020). Dialogues about emotional events between mothers with anxiety, depression, anorexia nervosa, and no diagnosis and their children. Parenting, 20(1), 69-82.

Response: The authors are grateful for the suggestion of R1. We agree that it is important to address intrapsychic changes related to the transition to motherhood, as well as the impact of mental health issues on child health and development. Accordingly, we have included these aspects in the manuscript based on the literature suggested by the reviewer.

In Introduction:

“Women face several intrapsychic challenges during the transition to motherhood, characterized by a "new psychic organization" where they redefine their roles and self-perceptions, often experiencing physical and psychological symptoms [5]. These symptoms linked to postpartum depression and anxiety indicate difficulties in adapting to the new roles [5]. Maternal distress-inducing factors also include changes in body, responsibilities, and social circumstances [6, 7]. Effective adaptation to motherhood involves internal resources (e.g., self-esteem and coping strategies) and external resources (e.g., social support), which mitigate this burden and protect mental health [7]. Notably, maternal mental health issues also affect their children. In this sense, when caregiver responsiveness is impaired, maternal-child interactions and attachment are disrupted [8, 9]. Consequently, children may exhibit poorer social engagement, regulatory behaviors, coping, emotionality, sleep quality, eating behavior, and development [10, 11]. Furthermore, long-term outcomes include an increased risk of psychosocial issues in adulthood [11]. Therefore, interventions to address maternal psychological disorders are crucial, as they improve both women and child health, mitigate long-term societal costs, and foster healthier emotional and developmental trajectories for children.”

In Conclusion:

“Research shows an association between maternal mental health issues and impaired child development, with both short- and long-term outcomes. Thus, our findings emphasize the critical importance of routine mental health assessments during perinatal visits, especially in crisis contexts. This highlights the need for targeted treatments and interventions for women at risk of postpartum depression, stress, and insomnia.”

2. Comment: Abstract: Acronyms for tools are used in the abstract. Authors are invited to eliminate acronyms. Specifically: "Postpartum depression (PDSS‐SF), insomnia (ISI), and perceived stress symptoms27 (PSS‐C) were used"

Response: Done.

3. Comment: Finally, the authors are invited to better clarify the hypotheses of the study, justifying them with the literature.

Response: The hypothesis of the study was included in the Introduction section.

Reviewer #2: The authors have done a good job with the organization of their article in general. To begin with, the research has been dealt with the technical aspect very well, and the conclusion has a very strong link that makes the article a very good one. In addition, the statistical aspect of the article is presented in a professional manner, although I have not gotten access to the original data to examine it. Furthermore, the authors have indicated that the data are available without restriction. Finally, the article presents the write-up in professional and appropriate English grammar throughout the article. I do recommend this article without any competing interest for publication.

Response: We greatly appreciate R2's comments. The data presented in this study are now openly available in Open Science Framework (OSF) at https://osf.io/v69tf/

General changes:

References: The following additional references were included to comply with the reviewers' comments:

5. Tambelli R, Ballarotto G, Trumello C, Babore A. Transition to Motherhood: A Study on the Association between Somatic Symptoms during Pregnancy and Post-Partum Anxiety and Depression Symptoms. Int J Environ Res Public Health. 2022;19(19):12861. doi: 10.3390/ijerph191912861

6. Rallis S, Skouteris H, McCabe M, Milgrom J. The transition to motherhood: towards a broader understanding of perinatal distress. Women Birth. 2014;27(1):68-71. doi: 10.1016/j.wombi.2013.12.004

7. Ben-Ari OT, Shlomo SB, Sivan E, Dolizki M. The transition to motherhood—A time for growth. Journal of Social and Clinical Psychology. 2009;28(8):943-70. doi: 10.1521/jscp.2009.28.8.943

8. Cimino S, Cerniglia L, Tambelli R, Ballarotto G, Erriu M, Paciello M, Oppenheim D, Koren-Karie N. Dialogues about emotional events between mothers with anxiety, depression, anorexia nervosa, and no diagnosis and their children. Parenting. 2020;20(1):69-82. doi: 10.1080/15295192.2019.1642688

9. Ballarotto G, Murray L, Bozicevic L, Marzilli E, Cerniglia L, Cimino S, Tambelli R. Parental sensitivity to toddler's need for autonomy: An empirical study on mother-toddler and father-toddler interactions during feeding and play. Infant Behav Dev. 2023;73:101892. doi: 10.1016/j.infbeh.2023.101892

10. Liu X, Wang S, Wang G. Prevalence and Risk Factors of Postpartum Depression in Women: A Systematic Review and Meta-analysis. J Clin Nurs. 2022;31(19-20):2665-2677. doi: 10.1111/jocn.16121

11. Hutchens BF, Kearney J. Risk Factors for Postpartum Depression: An Umbrella Review. J Midwifery Womens Health. 2020;65(1):96-108. doi: 10.1111/jm

---

## [Decision Letter · Decision Letter 1]

13 Sep 2024

PONE-D-23-34221R1Two-years mothering into the pandemic: Impact of the three COVID-19 waves in the Argentinian postpartum women’s mental healthPLOS ONE

Dear Dr. Miranda,

Thank you for submitting your manuscript to PLOS ONE. After careful consideration, we feel that it has merit but does not fully meet PLOS ONE’s publication criteria as it currently stands. Therefore, we invite you to submit a revised version of the manuscript that addresses the points raised during the review process.

We look forward to receiving your revised manuscript.

Kind regards,

Pracheth Raghuveer, MD, DNB

Academic Editor

PLOS ONE

**Journal Requirements:**

Reviewers' comments:

Reviewer's Responses to Questions

**Comments to the Author**

1. If the authors have adequately addressed your comments raised in a previous round of review and you feel that this manuscript is now acceptable for publication, you may indicate that here to bypass the “Comments to the Author” section, enter your conflict of interest statement in the “Confidential to Editor” section, and submit your "Accept" recommendation.

Reviewer #3: (No Response)

Reviewer #4: All comments have been addressed

2. Is the manuscript technically sound, and do the data support the conclusions?

Reviewer #3: Yes

Reviewer #4: Yes

3. Has the statistical analysis been performed appropriately and rigorously? 

Reviewer #3: Yes

Reviewer #4: Yes

4. Have the authors made all data underlying the findings in their manuscript fully available?

Reviewer #3: Yes

Reviewer #4: Yes

5. Is the manuscript presented in an intelligible fashion and written in standard English?

Reviewer #3: Yes

Reviewer #4: Yes

6. Review Comments to the Author

**Reviewer #3: ** the article is original, followed basic scientific reasoning with a systematic methodology especially the data analysis

**Reviewer #4:**  This is an excellent study and a well written paper. I have queries concerning PSS-C scores:

- PSS-C scores have not been presented for the first wave of the pandemic. Is there any specific reason for this lack of data? This should be mentioned in the methodology.

- As studying evolution of pandemic related stress was part of the test hypothesis, how did the authors deal with this missing data in their analysis? What possible effect could it have on the results, and conclusions?

7. PLOS authors have the option to publish the peer review history of their article (what does this mean? ). If published, this will include your full peer review and any attached files.

**Do you want your identity to be public for this peer review?** For information about this choice, including consent withdrawal, please see our Privacy Policy .

Reviewer #3: **Yes: ** Dr. Abioye Opeyemi Oladipupo

Reviewer #4: No

---

## [Author Response · Author response to Decision Letter 2]

16 Sep 2024

Dear Editor,

We thank the reviewer for the insightful suggestions, which have greatly improved our work. The changes made are listed below:

• Comment: The abstract should adjusted for clarity especially on the concerned factors, such can be named negative factors affecting mental health.(line 31)

Response: The sentence has been rephrased for clarity.

• Comment: ‘No medical support’ can be changed to ‘lack of medical support’ (line 31)

Response: ‘No medical support’ was replaced by ‘lack of medical support’.

• Comment: Line 38 to 40 in abstract can be rephrased to “Thus, health systems must see to protection of women of reproductive age group against negative factors in order to cope with pandemic-related events”.

Response: The sentence has been rephrased.

• Comment: The introduction can be improved by mentioning pre-Covid 19 data on postpartum depression briefly to expatiate on the disease burden.

Response: This issue has been addressed in the Discussion section (paragraph 3, lines 381-383). Also, the pre-COVID prevalence of maternal mental health issues has been addressed in the Introduction section (paragraph 3, lines 76-79).

• Comment: Methodology is adequate. Type of lactation should be changed to type of feeding since mixed feeding and artificial formula were part of the options.

Response: “Type of lactation” has been replaced by “type of feeding”.

• Comment: Conclusion can be trimmed down to few details on the findings and recommendations against future similar epidemiologic scenarios.

Response: Conclusion section has been revised and re-written to focus more concisely on the key findings and recommendations for future similar epidemiologic scenarios, as follows: “In summary, our results demonstrate that the mental health of Argentinian puerperal women deteriorated during the COVID-19 pandemic. Our research findings suggest a notable increase in the prevalence of postpartum depression and insomnia during the outbreaks. Moreover, the social and economic impact involved a high psychological risk, although women have become able to cope with pandemic-related stress. Thus, these findings emphasize the critical importance of routine mental health assessments during perinatal visits, especially in future similar epidemiologic scenarios, to address the risk of postpartum depression, stress, and insomnia for vulnerable mothers.”

The authors hope that this revised version meets the standards for publication in PLOS ONE.

Respectfully,

Agustin Ramiro Miranda

Institut de Recherche pour le Développement (IRD)

---

## [Decision Letter · Decision Letter 2]

27 Dec 2024

PONE-D-23-34221R2Two-years mothering into the pandemic: Impact of the three COVID-19 waves in the Argentinian postpartum women’s mental healthPLOS ONE

Dear Dr. 

Thank you for submitting your manuscript to PLOS ONE. After careful consideration, we feel that it has merit but does not fully meet PLOS ONE’s publication criteria as it currently stands. Therefore, we invite you to submit a revised version of the manuscript that addresses the points raised during the review process.

We look forward to receiving your revised manuscript.

Kind regards,

Pracheth Raghuveer, MD, DNB

Academic Editor

PLOS ONE

Journal Requirements:

Reviewers' comments:

Reviewer's Responses to Questions

**Comments to the Author**

1. If the authors have adequately addressed your comments raised in a previous round of review and you feel that this manuscript is now acceptable for publication, you may indicate that here to bypass the “Comments to the Author” section, enter your conflict of interest statement in the “Confidential to Editor” section, and submit your "Accept" recommendation.

Reviewer #4: (No Response)

2. Is the manuscript technically sound, and do the data support the conclusions?

Reviewer #4: Yes

3. Has the statistical analysis been performed appropriately and rigorously? 

Reviewer #4: Yes

4. Have the authors made all data underlying the findings in their manuscript fully available?

Reviewer #4: No

5. Is the manuscript presented in an intelligible fashion and written in standard English?

Reviewer #4: Yes

6. Review Comments to the Author

Reviewer #4: The manuscript does not address the comments I had made in the previous round of review. I have copied them here:

This is an excellent study and a well written paper. I have queries concerning PSS-C scores:

- PSS-C scores have not been presented for the first wave of the pandemic. Is there any specific reason for this lack of data? This should be mentioned in the methodology.

- As studying evolution of pandemic related stress was part of the test hypothesis, how did the authors deal with this missing data in their analysis? What possible effect could it have on the results, and conclusions?

7. PLOS authors have the option to publish the peer review history of their article (what does this mean? ). If published, this will include your full peer review and any attached files.

**Do you want your identity to be public for this peer review?** For information about this choice, including consent withdrawal, please see our Privacy Policy .

Reviewer #4: No

---

## [Author Response · Author response to Decision Letter 3]

22 Jan 2025

Dear Editor,

We appreciate the valuable suggestions provided by the reviewer, which have enhanced our work. Consequently, each change made is indicated in the following list:

Comment 1: Have the authors made all data underlying the findings in their manuscript fully available? Reviewer #4: No

Response: As addressed in a previous round, the data presented in this study are openly available in Open Science Framework (OSF) at https://osf.io/v69tf/files/osfstorage

Comment 2: The manuscript does not address the comments I had made in the previous round of review. I have copied them here: This is an excellent study and a well written paper.

I have queries concerning PSS-C scores:

- PSS-C scores have not been presented for the first wave of the pandemic. Is there any specific reason for this lack of data? This should be mentioned in the methodology.

Response: Thank you for your feedback and for bringing this to our attention. We sincerely apologize for the oversight and any confusion caused. The comments from the previous round of review were inadvertently not addressed in the revised manuscript. This was an unintentional error, and we regret that it was missed. We have now carefully reviewed your previous comments and have made the necessary revisions to ensure that all of your suggestions are properly incorporated. The updated manuscript reflects these changes, and we believe they improve the overall quality of the study.

The specific reason for the lack of PSS-C data during the first wave of COVID-19 is that the validation of the instrument was published after this period. This is understandable, as the authors who adapted the PSS to the pandemic context conducted their research at the beginning of the pandemic, with their manuscript being published in July 2020, the same month when data collection for the first sample in our study was completed. This has been addressed in Discussion section as stated in the next response.

- As studying evolution of pandemic related stress was part of the test hypothesis, how did the authors deal with this missing data in their analysis? What possible effect could it have on the results, and conclusions?

Response: The following text has been added in Discussion section: “Second, as part of our exploration of the evolution of pandemic-related stress, the absence of PSS-C data during the first wave of COVID-19 represents a potential limitation. This gap arose because the Spanish PSS-C adaptation for the pandemic context was published after data collection for our initial sample had been completed [84]. Furthermore, since the lack of data affected all participants in this wave, rather than a specific subset, we opted not to impute the missing values in order to prevent potential bias. Considering the exceptional circumstances of the first wave, which was marked by unprecedented levels of uncertainty and stress for all, and recognizing that this phase tended to be the most stressful [85], the absence of data during this period is not expected to substantially affect the overall conclusions of the study.”.

General changes:

The following additional references were included to comply with the reviewer's comments:

84. Campo-Arias A, Pedrozo-Cortés MJ, Pedrozo-Pupo JC. Pandemic-Related Perceived Stress Scale of COVID-19: An exploration of online psychometric performance. Rev Colomb Psiquiatr. 2020;49:229-230. English, Spanish. doi: 10.1016/j.rcp.2020.05.005.

85. Bendau A, Asselmann E, Plag J, Petzold MB, Ströhle A. 1.5 years pandemic - Psychological burden over the course of the COVID-19 pandemic in Germany: A nine-wave longitudinal community study. J Affect Disord. 2022;319:381-387. doi: 10.1016/j.jad.2022.09.

All issues raised by the reviewer have been addressed. Thus, the authors hope that this article will be appropriate for Plos One.

Respectfully,

Agustin Ramiro Miranda

Institut de Recherche pour le Développement (IRD)

---

## [Editor Report · Decision Letter 3]

18 Feb 2025

Two-years mothering into the pandemic: Impact of the three COVID-19 waves in the Argentinian postpartum women’s mental health

PONE-D-23-34221R3

Dear Dr. Miranda,

We’re pleased to inform you that your manuscript has been judged scientifically suitable for publication and will be formally accepted for publication once it meets all outstanding technical requirements.

Kind regards,

Pracheth Raghuveer, MD, DNB

Academic Editor

PLOS ONE
---

## [Editor Report · Acceptance letter]

PONE-D-23-34221R3

PLOS ONE

Dear Dr. Miranda,

I'm pleased to inform you that your manuscript has been deemed suitable for publication in PLOS ONE. Congratulations! Your manuscript is now being handed over to our production team.

Kind regards,

on behalf of

Dr. Pracheth Raghuveer

Academic Editor

PLOS ONE